# RCKD: Response-Based Cross-Task Knowledge Distillation for Pathological Image Analysis

**DOI:** 10.3390/bioengineering10111279

**Published:** 2023-11-02

**Authors:** Hyunil Kim, Tae-Yeong Kwak, Hyeyoon Chang, Sun Woo Kim, Injung Kim

**Affiliations:** 1Deep Bio Inc., Seoul 08380, Republic of Korea; hikim@deepbio.co.kr (H.K.); tykwak@deepbio.co.kr (T.-Y.K.); hychang@deepbio.co.kr (H.C.); swkim@deepbio.co.kr (S.W.K.); 2School of Computer Science and Electrical Engineering, Handong Global University, Pohang 37554, Republic of Korea

**Keywords:** deep learning, nuclei segmentation, knowledge distillation, contrastive learning, self supervised learning

## Abstract

We propose a novel transfer learning framework for pathological image analysis, the Response-based Cross-task Knowledge Distillation (RCKD), which improves the performance of the model by pretraining it on a large unlabeled dataset guided by a high-performance teacher model. RCKD first pretrains a student model to predict the nuclei segmentation results of the teacher model for unlabeled pathological images, and then fine-tunes the pretrained model for the downstream tasks, such as organ cancer sub-type classification and cancer region segmentation, using relatively small target datasets. Unlike conventional knowledge distillation, RCKD does not require that the target tasks of the teacher and student models be the same. Moreover, unlike conventional transfer learning, RCKD can transfer knowledge between models with different architectures. In addition, we propose a lightweight architecture, the Convolutional neural network with Spatial Attention by Transformers (CSAT), for processing high-resolution pathological images with limited memory and computation. CSAT exhibited a top-1 accuracy of 78.6% on ImageNet with only 3M parameters and 1.08 G multiply-accumulate (MAC) operations. When pretrained by RCKD, CSAT exhibited average classification and segmentation accuracies of 94.2% and 0.673 mIoU on six pathological image datasets, which is 4% and 0.043 mIoU higher than EfficientNet-B0, and 7.4% and 0.006 mIoU higher than ConvNextV2-Atto pretrained on ImageNet, respectively.

## 1. Introduction

### 1.1. Background

Pathological image analysis aims to extract useful information from pathological images commonly acquired through a whole slide scanner or camera. It covers various tasks, such as classification, segmentation, and detection of cells, nuclei, or cancerous regions, and is one of the core technologies for computer-aided diagnosis. Since deep learning exhibited outstanding performance in the ImageNet challenge [1], researchers have actively applied deep learning to pathological image analysis. Deep learning showed excellent performance in multiple challenges such as mitosis detection [2,3], breast cancer classification [4], and gland segmentation [5]. Currently, deep learning is widely used as a core algorithm in many pathological image analysis challenges [6]. In spite of that, there is a lot of room for improvement.

One of the main challenges is the difficulty of building a large-scale dataset for each pathological image analysis task. Collecting a large amount of data is restricted by privacy concerns. Moreover, labeling pathological images is more difficult and expensive than ordinary images. For example, training a deep learning model that predicts the aggressiveness of a tumor requires a dataset containing mitosis counting, cell segmentation labels, and the patient’s prognosis. Building such a dataset consumes significant time and effort from skilled pathologists [7].

Transfer learning is widely used in pathological image analysis to overcome the scarcity of labeled data and achieve high performance. A widely used approach is the pretraining and fine-tuning strategy, which first learns general knowledge by pretraining a model for an upstream task that can share knowledge with the target task using a large dataset, and then fine-tunes the pretrained model for the target task, also called the downstream task, using a relatively small target dataset [8,9]. However, because the characteristics of pathological images differ significantly from those of general images, transferring knowledge learned from a general image dataset, such as ImageNet [10], to a pathological image analysis task is less effective than ordinary transfer learning settings.

As an alternative, researchers have actively studied to learn features from unlabeled pathological images [11,12,13,14,15,16,17]. Recently, various unsupervised and self-supervised learning techniques, such as contrastive learning and masked autoencoders, have exhibited excellent performance in many computer vision tasks [18]. However, self-supervised learning algorithms do not perform as well in pathological image analysis as it does in other areas of computer vision, as described in Section 1.2.1. As a result, none of the existing supervised or self-supervised pretraining methods showed sufficient performance in pathological image analysis.

In addition, pathological images generally have significantly higher resolution than ordinary images, resulting in substantial increases in computational and memory requirements. For example, The Cancer Genome Atlas (TCGA) dataset [19] consists of images with a resolution of 20,000×40,000 pixels, which is hundreds of times the size of images in conventional datasets. Most pathological image analysis models decompose the whole-slide images (WSI) into patches to reduce the overhead. Nevertheless, memory and computational load is still an important issue in pathological image analysis. Many studies have been conducted to reduce deep learning models, but more research is needed to process high-resolution pathological images. Consequently, to achieve high performance in pathological image analysis in a general computing environment, we need not only a pretraining method that is effective in learning knowledge from unlabeled pathological images but also a lightweight architecture to reduce computational and memory overhead.

### 1.2. Related Work

In this subsection, we briefly introduce prior studies on self-supervised learning for pathological image analysis and efficient network architectures. We also present prior work on knowledge distillation and visual attention models on which the proposed methods are based.

#### 1.2.1. Self-Supervised Learning for Pathological Image Analysis

Inspired by the success of self-supervised learning (SSL) in computer vision, many researchers have applied these techniques to pathological image analysis. Boyd et al. [11] applied a generative model-based learning method called visual field expansion. Ciga et al. [12] and Dehaene et al. [13] applied contrastive learning techniques such as SimCLR [20] and MoCoV2 [21] to pathological image analysis.

However, existing self-supervised pretraining techniques are less effective in pathological image analysis than in other computer vision fields. Zhang et al. [14] and Li et al. [15] reported analysis results suggesting that contrastive learning techniques are less effective for pathological image analysis. Koohbanani et al. [16] show a significant difference in performance between domain-agnostic and domain-specific tasks, suggesting that pathological images should be analyzed using pathological image-specific learning methods. Lin et al. [17] suggest a method to improve the performance of contrastive learning on pathological images by increasing the self-invariance, intra-invariance within a WSI, and inter-invariance across WSIs of the feature. However, their method requires clustering the feature vectors in each epoch, resulting in a significant increase in memory requirements.

#### 1.2.2. Knowledge Distillation

Knowledge distillation (KD) is a technique for improving the training of a relatively small student model by exploiting the knowledge of a large and powerful teacher model. KD has been widely used to compress heavy models or to improve the performance of lightweight models. In the early days of KD, the student model learned to mimic the logits of the teacher model for each training sample [22], which is called response-based knowledge distillation (RKD). In a subsequent study, Adriana et al. [23] proposed feature-based knowledge distillation, which trains the student model using the intermediate features of the teacher model, enabling a more accurate approximation of the teacher model [24]. However, KD is primarily used under the assumption that the teacher and student models perform the same task in the same domain. KD performance often severely decreases when the task or domain of the student model differs from that of the teacher model. Li et al. [25] argued that the limitation comes from the fact that KD mainly transfers knowledge about global representation, and KD is less effective in transferring knowledge about local representation. To address this problem, they suggested using sub-modules to complement local knowledge.

A few previous studies apply KD to pathological image analysis. DiPalma et al. [26] propose a method for improving the computational efficiency of applying KD between models with the same structure but different input resolutions, and Javed et al. [27] proposed using additional modules to transfer knowledge stage-by-stage to learn robust tissue heterogeneity. Zhang et al. [28] presented a method of distilling knowledge from teacher models trained on diverse pathological datasets to help the student model learn the characteristic features of various pathological images.

However, these studies aim to reduce the size of a high-performance model that already exists and do not improve the performance of the high-performance model. It is hard to improve the performance of a model for a pathological image analysis task through conventional KD. Since the student model distills knowledge from the teacher model, a strong teacher model is a key requirement for KD. In pathological image analysis, where building a large-scale dataset is difficult, it is challenging to build a teacher model that performs the same task as the student model and is powerful enough to guide the learning of the student model.

#### 1.2.3. Efficient Network Architectures for Image Analysis

There is a large body of previous work on the design of efficient network architectures. VGG16 [29] demonstrated that stacking multiple convolution filters with a kernel size of 3×3 can approximate a large kernel in image classification tasks. SqueezeNet [30] reduced the number of parameters using 1×1 convolutions and squeeze-expand modules, achieving similar performance to AlexNet with 50× fewer parameters. ResNet [31] proposed a bottleneck structure to reduce the number of parameters in convolution blocks. MobileNet [32] reduced the number of parameters by up to 11% by decomposing convolution operations into a depthwise convolution and a pointwise convolution.

MobileNetV2 [33] achieved outstanding performance with a small number of parameters and computation using an inverted residual block and a linear bottleneck. Most of the recent architectures based on convolution or self-attention, such as EfficientNet [34], EfficientNetV2 [35], CoAtNet [36], ConvNext [37], ConvNextV2 [38], EfficientFormer [39], and EfficientFormerV2 [40], that exhibited good performance on ImageNet adopt the MBConv module proposed in [33]. In particular, CoAtNet proposes an architecture that combines CNN and self-attention. ConvNext achieved higher performance than Swin-B [41] of similar size by modernizing ResNet with several recent techniques such as the AdamW optimizer, a patchfy stem, large kernels of 7×7 size, fewer activation functions, and layer normalization.

#### 1.2.4. Visual Attention

The attention mechanism allows the model to concentrate on essential features by assigning a weight to each feature based on its relevance. In general image processing, attention mechanisms are often categorized into channel attention, which learns the feature type to focus on, and spatial attention, which learns the locations of important features. Different channels of a feature map in a deep neural network represent different objects or concepts [42]. Hu et al. [43] proposed SENet, which estimates the importance of each channel and scales the channels accordingly. Gao et al. [44] pointed out that SENet is a simple structure designed to focus on important global information, so a more sophisticated structure is needed to focus on details better. Lee et al. [45] further reduced the size of SENet by applying a lightweight channel-wise fully-connected (CFC) layer. However, channel attention models can only learn what to focus on, but not where. Chen et al. [42], Park et al. [46], Woo et al. [47] demonstrated that using a combination of spatial attention and channel attention is superior to using channel attention alone. Wang et al. [48] showed that utilizing self-attention-based spatial attention can improve the performance of CNNs.

Dosovitskiy et al. [49] propose a vision Transformer (ViT), which modifies the Transformer network to fit image processing and achieves higher performance than CNN on the ImageNet dataset for the first time. Subsequent studies propose various image processing models based on self-attention, such as SwinTransformer [41], which extends ViT to a multi-scale structure, and CoAtNet, which combines separable convolution and multi-head self-attention (MSA).

### 1.3. Research Objective

In this study, we aim to develop a pretraining method and a lightweight network architecture to overcome the aforementioned challenges and improve the performance and efficiency of various pathological image analysis tasks. To this end, we propose the Response-based Cross-task Knowledge Distillation (RCKD) framework to learn knowledge from unlabeled pathological images using a high-performance model developed for a different task. We also propose the Convolutional neural network with Spatial Attention by Transformer (CSAT), an effective and efficient architecture designed as a backbone network for high-resolution image analysis. CSAT integrates multiple techniques that have shown to be effective in recent studies on lightweight architectures and further improves the performance through a novel Spatial Attention by Transformer (SAT) module. We expect that the results of this study will help reduce the cost of pathological image analysis and improve diagnostic performance by providing pathologists and researchers with fast and efficient diagnostic and research methods.

## 2. Materials and Methods

### 2.1. Datasets

In this study, we used different datasets for three specific purposes. For the pretraining of the model, we used the TCGA dataset, which consists of a large number of high-resolution pathological images. For fine-tuning and evaluation of the downstream tasks, we used six datasets described in Section 2.1.2. In addition, we used the ImageNet dataset to evaluate the performance and efficiency of network architectures in the analysis of general images not limited to pathological images.

#### 2.1.1. Pretraining Dataset

TCGA dataset is one of the largest publicly available cancer genome datasets collected primarily for use in the diagnosis, treatment, and prevention of cancer. TCGA dataset includes more than 20,000 WSIs of stained tissue samples that belong to 33 different cancer types. TCGA dataset is widely used in various fields of pathological image analysis. From the TCGA dataset, we collected 11,716 WSIs from 32 different types of cancer. We excluded the formalin-fixed paraffin-embedded (FFPE) slide images because their quality was poor. These samples were collected from a variety of organs, including the breast, brain, ovary, lung, kidney, prostate, stomach, and liver. Then, we cut the selected WSIs into non-overlapping patches of 1024×1024 size at 20× magnification. We removed patches whose average intensity values are not in the range of [50,245] because most of such patches consist of backgrounds rather than tissues. In this way, we collected 9,229,324 tissue patches. The types of studies and the number of whole slide imaging (WSI) are presented in Table A1. Among them, we randomly selected 400,000 patches and used them for pretraining.

#### 2.1.2. Downstream Tasks and Datasets

For the fine-tuning and evaluation of the pretrained models, we used four segmentation datasets and two segmentation datasets. For the classification tasks, we used BACH (microscopy) [50], CRC [51], BreakHis [52], and  Lymph [53] datasets.

**The breast cancer histology images (BACH) microscopy dataset** contains 400 hematoxylin and eosin (H&E) stained microscopy image patches categorized into four classes: normal, benign, in situ carcinoma, and invasive carcinoma. Each class has 100 training image patches. The average patch size is 2048×1536 pixels.**The colorectal cancer (CRC) dataset** was collected for the classification of tissue areas from H&E stained WSIs of colorectal adenocarcinoma patients. It includes a total of 100,000 non-overlapping image patches in the training dataset extracted from 86 WSIs of cancer and normal tissues. The average patch size is 224×224 pixels. The patches were manually labeled by pathologists into nine tissue classes: adipose (ADI), background (BACK), debris (DEB), lymphocytes (LYM), mucus (MUC), smooth muscle (MUS), normal colon mucosa (NORM), cancer-associated stroma (STR), and colorectal adenocarcinoma epithelium (TUM). The validation dataset includes 7180 image patches extracted from 50 colorectal adenocarcinoma patients.**The breast cancer histopathological (BreakHis) dataset** was collected for binary breast cancer classification. It contains a total of 7909 breast tumor tissue image patches from 82 patients, consisting of 2480 benign and 5429 malignant tumor patches. BreakHis includes four types of benign tumors: adenosis, fibroadenoma, phyllodes tumor, and tubular adenoma, as well as four types of malignant tumors: ductal carcinoma, lobular carcinoma, mucinous carcinoma, and papillary carcinoma. The average patch size is 700×460 pixels.**The Lymph dataset** was collected for the classification of malignant lymph node cancer. It provides a total of 374 training image patches, with 113 image patches of chronic lymphocytic leukemia (CLL), 139 image patches of follicular lymphoma (FL), and 122 image patches of mantle cell lymphoma (MCL). The average patch size is 1388×1040 pixels.

For the segmentation tasks, we used the BACH (WSI) [50] and GlaS [54] datasets.

**The BACH WSI dataset** consists of 10 WSIs with an average size of 40,517×58,509 scanned at a resolution of 0.467 µ/pixel. It includes pixel-wise annotations in four region categories (normal, benign, in situ carcinoma, and invasive carcinoma). In this study, we cut the WSIs into 4,453 non-overlapping patches of 1024×1024 size at 10× magnification.**The gland segmentation in colon histology images (GlaS) dataset** is the benchmark dataset for the Gland Segmentation Challenge Contest at MICCAI in 2015. It consists of 165 image patches derived from 16 H&E stained histological sections of stage T3 or T42 colorectal adenocarcinoma. Each section originates from a different patient. The WSIs were digitized at a pixel resolution of 0.465 µm. The GlaS dataset consists of 74 benign and 91 malignant gland image patches, each with pixel-wise annotations. The average patch size is 775×522 pixels.

Since the BACH (microscopy), BreakHis, and Lymph datasets only release training sets to the public, we randomly split their training sets by 6:2:2 and used them for training, validation, and testing, respectively. The CRC and GlaS datasets provide the training and validation sets but not the test set. We used 80% of their training sets for training and 20% for validation and measured the performance on the validation set. The BACH (WSI) dataset consists of ten WSIs. We used five of them for training, three for validation, and two for testing. Table 1 summarizes the task, classes, data size, number of patches, magnification ratio, and size of patches for the downstream datasets.

#### 2.1.3. General Image Dataset

We used the ImageNet dataset to evaluate the performance and efficiency of model architectures in general image classification. The ImageNet large-scale visual recognition dataset includes 1000 classes, ranging from common objects such as ‘banana’ to abstract concepts like ‘bubble’. It contains 1,281,167 training images, 50,000 validation images, and 100,000 test images. The average image size within the dataset is approximately 469×387 pixels.

### 2.2. Response-Based Cross-Task Knowledge Distillation

The pretraining and fine-tuning strategy is a widely used approach to achieve high performance with relatively small datasets. However, existing pretraining methods are ineffective for pathological images, as described in the previous section. To overcome this limitation, we repurpose knowledge distillation to learn knowledge from unlabeled pathological images. RCKD is a novel transfer learning framework for pathological image analysis, which can be used as an alternative to the existing pretraining techniques.

Figure 1 illustrates the overall procedure of RCKD. The RCKD framework pretrains the backbone of a pathological image analysis model as a student model with the guidance of a high-performance teacher model. The teacher model is a high-performance network developed for a different task, such as nuclei segmentation, that can share knowledge with pathological image analysis tasks. First, the teacher model predicts binary nuclei segmentation maps of unlabeled pathological images. Then, the student model that combines the backbone and a segmentation head learns to output the binary nuclei segmentation map as close as possible to the output of the teacher model. At the same time as it is learning nuclei segmentation with the prediction of the teacher model as a pseudo label, the student model learns features useful for pathological image analysis. After pretraining, we remove the nuclei segmentation head from the student model, combine the pretrained backbone with a new head for the downstream task, and fine-tune it using the target data. The detailed procedure is described in Section 2.2.1 and Section 2.2.2.

Nuclei segmentation is an effective upstream task to pretrain pathological image analysis models for the following reasons. First, pathological images are generally composed of textures rather than large-scale objects with fixed shapes. Therefore, learning low-level local features is crucial in most pathological image analysis tasks. Since nuclei segmentation is a pixel-level classification task, it drives the model to learn local and positional information. Second, a cell is the basic unit of an organ [55], and the nucleus is the core component of a cell. Most pathological images contain many nuclei and can, therefore, be used as training data for nuclei segmentation. Third, since nuclei have less shape variation than other components in pathological images, the features learned for nuclei segmentation can be useful for analyzing other types of pathological images. Fourth, the state-of-the-art (SOTA) nuclei segmentation models provide excellent performance and are sufficient to guide the pretraining of the student model.

#### 2.2.1. Pretraining from Unlabeled Pathological Images

The teacher model takes an unlabeled pathological image *x* as input and predicts a binary nuclei segmentation map as Equation (Equation 1).
(1)$$y=f_{teacher}\left(x\right),\;for\;x\in D,$$
where $f_{teacher}\left(\cdot \right)$ is a teacher model, $D\subset R^{3xHxW}$ is a set of pathological images without segmentation labels, and $y\in R^{HxW}$ is a probability map predicted by $f_{teacher}\left(\cdot \right)$, where $y_{ij}$ is the estimated probability that a pixel $x_{ij}$ belongs to a nuclei region. Then, we convert *y* into a binary segmentation map $N\left(x\right)\in {\left\{0,1\right\}}^{HxW}$ with a threshold value α, as Equation (Equation 2).
(2)$$N{\left(x\right)}_{ij}=\left\{\begin{array}{c} 1\;\mathrm{if}\;y_{ij}>=\alpha \\ 0\;\mathrm{otherwise}, \end{array}\right.$$
where α=0.5 in our study.

For the teacher model $f_{teacher}\left(\cdot \right)$, we used StarDist [56] pretrained on the MoNuSeg2018 [57] and TNBC [58] datasets. StarDist won the CoNIC challenge in 2022 [59]. StarDist segments nuclei regions using a U-Net [60] based model and represents them as star convex polygons. The structure and hyperparameters of StarDist are presented in Figure 2. Figure 3 displays the pathological image samples used for pretraining and the pseudo label N(x) estimated by StarDist.

To pretrain the backbone network via RCKD, we combine it with a segmentation head and train the combined model to predict the segmentation map identical to the pseudo label N(x) for each image x∈D. In this study, we implemented the backbone network using a novel lightweight network, CSAT, described in Section 2.3, and the segmentation head using the U-Net decoder. The pretraining loss $L_{KD}\left(\theta |x\right)$ is defined as Equation (Equation 3).
(3)$$\begin{array}{cc} L_{KD}\left(\theta |x\right) & =\frac{1}{H\times W}\sum_{i}^{H}\sum_{j}^{W}CE\left(f_{student}\left(x_{ij};\theta \right),N\left(x_{ij}\right)\right),\\ \theta & ={\mathrm{argmin}}_{\theta }\frac{1}{\left|D\right|}\sum_{x\in D}L_{KD}\left(\theta |x\right) \end{array}$$
where $f_{student}\left(x;\theta \right)$ is a student model parameterized by θ and CE(·,·) denotes the cross-entropy loss. After pretraining, we transfer the student model $f_{student}\left(x;\theta \right)$, replace the nuclei segmentation head with a new head for the target task, and fine-tune it on the target data.

Following the unified scheme for ImageNet (USI) [61], we first pretrain the student model on ImageNet data to improve performance and training speed before RCKD. We optimize the student model using the layer-wise adaptive rate scaling (LARS) optimizer [62] with an initial learning rate of zero and a batch size of 64. LARS automatically adjusts the learning rate for each layer, allowing training to proceed even when the initial learning rate is set to zero. We normalize the input image to 512×512 size by bilinear interpolation and warm up for 10 epochs out of 100 training epochs. The pretraining procedure is summarized in Algorithm  1.

**Algorithm 1** Pretraining Procedure1: *W*▹ WSI in pretraining dataset (TCGA)2: KD▹ Knowledge distillation3: $f_{teacher}$▹ Teacher model4: $f_{student}$▹ Student model5: Mag▹ Magnification ratio of WSI6: $P_{size}$▹ Patch size7: $S_{n}$▹ Number of samples for pretraining8: **procedure** Pretrain($W,KD,f_{teacher},f_{student},Mag,P_{size},S_{n}$)9:      $P=GetPatches(W,Mag,\left(P_{size},P_{size}\right))$▹ Get patches from a WSI10:    $P_{tissue}=SelectTissuePatches\left(P\right)$▹ Discard background patches11:    $D=RandomSampling(P_{tissue},S_{n})$▹ Select patches for pretraining12:    **for** all *x* in *D* **do**13:          $N=f_{teacher}\left(x\right)$▹ Make pseudo label14:          $Loss=KD(f_{student}\left(x\right),N)$▹ Knowledge distillation15:          $Backpropagate(Loss,f_{student})$16:    **return** $f_{student}$* In this study, Mag=20, $P_{size}=1024$, and $S_{n}=400,000$.

#### 2.2.2. Fine-Tuning for Pathological Image Analysis Tasks

To transfer the knowledge of the pretrained student model to the downstream task, we first build a model $f_{target}\left(x;\theta \right)$ for the target task by replacing the nuclei segmentation head used to pretrain the student model with a new head for the target task. Then, we fine-tune $f_{target}\left(x;\theta \right)$ by supervised learning to minimize a task-specific loss on the labeled target dataset. In this study, we fine-tuned $f_{target}\left(x;\theta \right)$ to four classification datasets: BACH (microscopy), CRC, BreakHis, Lymph, as well as two segmentation datasets: BACH (WSI), GlaS. We used focal loss [63] to mitigate potential problems caused by data imbalance.

The model for a classification task predicts the probability that the input image belongs to each class in the form of a vector as Equation (Equation 4).
(4)$$\begin{array}{c} y_{s}\left(x\right)=f_{target}\left(x;\theta \right)\in R^{T}, \end{array}$$
where *T* is the number of target classes. We fine-tune the parameter θ to minimize the focal loss defined as Equation (Equation 5).
(5)$$\begin{array}{c} L_{classification}\left(\theta |x\right)=-\alpha {\left(1-y_{s}{\left(x\right)}_{t}\right)}^{\gamma }\log y_{s}{\left(x\right)}_{t}, \end{array}$$
where $\left(x,t\right)\in D_{target}$ is a pair of target data, and its class label and parameters α and γ are set to 0.25 and 2, respectively.

A segmentation model predicts the probability of each target class (e.g., an object or region) for each pixel in the form of a 3D map as Equation (Equation 6).
(6)$$\begin{array}{c} y_{s}\left(x\right)=f_{target}\left(x;\theta \right)\in R^{T\times H\times W}, \end{array}$$
where H×W is the size of the input image, and *T* is the number of target objects or regions. The loss function for the fine-tuning of a segmentation task is defined as Equation (Equation 7).
(7)$$\begin{array}{c} L_{segmentation}\left(\theta |x\right)=-\frac{1}{HxW}\sum_{i}^{H}\sum_{j}^{W}\alpha {\left(1-y_{s}{\left(x\right)}_{tij}\right)}^{\gamma }\log y_{s}{\left(x\right)}_{tij} \end{array}$$

When fine-tuning the model for classification tasks, we implemented classification heads by combining a linear layer and a softmax layer. For segmentation tasks, we built segmentation heads following the decoder of U-Net. To reduce the potential bias caused by the composition of the training, validation, and test sets, which can be serious in pathological image analysis where the number of samples is usually small, we repeated the data splitting and experiment five times by changing the random seed and evaluated the models by the average performance. The fine-tuning procedure is summarized in Algorithm  2.

**Algorithm 2** Fine-tuning Procedure1: $f_{target}$▹ Pretrained student model2: *D*▹ Target dataset3: $N_{fold}$▹ Total number of folds4: $N_{epoch}$▹ Total number of training epochs5: $N_{stop}$▹ Patience number for early stopping6: **procedure** Fine-tune($f_{target},D,N_{fold},N_{epoch},N_{stop}$)7:       **for** *fold* in range [1, $N_{fold}$] **do**8:             $X_{train},Y_{train},X_{val},Y_{val},X_{test},Y_{test}$ = LoadData(D,fold)9:             **for** epoch in range [1, $N_{epoch}$] **do**10:                 **for** *i* in range [1, $\left|X_{train}\right|$] **do**11:                       $L_{train}=L_{focal}\left(f_{target}\left(X_{train}\left(i\right)\right),Y_{train}\left(i\right)\right)$12:                       Backpropagate$\left(L_{train},f_{target}\right)$13:                 $L_{Val}=\frac{1}{\left|X_{val}|\right)}\sum_{j=1}^{\left|X_{val}\right|}L_{focal}(f_{target}\left(X_{val}\left(j\right),Y_{val}\left(j\right)\right)$14:                 CheckForEarlyStopping($L_{val},N_{stop}$)15:           $A_{test}=\frac{1}{\left|X_{test}\right|}\sum_{k=1}^{\left|X_{test}\right|}$Accuracy$\left(f_{target}\left(X_{test}\left(k\right)\right),Y_{test}\left(k\right)\right)$16:      $A_{average}$ = $\frac{1}{N_{fold}}\sum_{fold=1}^{N_{fold}}A_{test}\left(fold\right)$17:      **return $A_{average}$*** In this study, $N_{fold}=5$, $N_{epoch}=200$, and $N_{stop}=20$.

### 2.3. Convolutional Neural Network with Spatial Attention by Transformer

In this section, we present CSAT, a lightweight network designed for use as a general-purpose backbone network in high-resolution pathological image analysis. CSAT combines multiple techniques that have been proven effective in recent studies on lightweight networks. In addition, we improved its performance by adding a novel SAT module. Moreover, we reduce the computational and memory requirements of the Transformer for computing spatial attention by estimating attention weights at a reduced resolution and then upsampling the attention map to the size of the feature map.

#### 2.3.1. Overall Structure

CoAtNet and AlterNet [64] exhibited improved performance by applying convolution for feature extraction in the low-level layers and aggregating features using multi-head self-attention (MSA) or Transformer blocks in the high-level layers. A few subsequent models, such as EfficientFormerV2, apply similar network configurations. CSAT also applies convolutions to extract features in the low-level layers and Transformers to aggregate feature maps in the high-level layers. The overall structure of CSAT is illustrated in Figure 4.

The structure and hyper-parameters of CSAT are based on EfficientFormerV2, a lightweight network designed for mobile environments. The bottom of CSAT consists of a patchfy stem [37] that splits the image into patches and reshapes each patch into a vector. These patches are then fed into the subsequent stages. Stages 1 and 2 are composed of two SAT blocks described in Section 2.3.2. Stages 3 and 4 consist of six and four SAT blocks followed by two Transformer blocks [65]. The SAT block extracts local features by convolutions and then re-scales the feature elements by spatial attention computed by an SAT module. The structure and hyper-parameters of CSAT are presented in Table 2.

#### 2.3.2. SAT Block

We designed the front part of the SAT block following Woo et al. [38]. It first extracts local features using a depthwise convolution followed by a 1 × 1 convolution as in Howard et al. [32]. Then, it applies global response normalization (GRN) to prevent any particular feature map from being overly dominant. It also applies an additional 1×1 convolution to abstract the feature map.

The rear part consists of a SAT module. As described in Section 2.3.3, the SAT module re-scales the features by multiplying them by attention weights estimated from the global context via a Transformer. Finally, the SAT block adds the input feature map, as in He et al. [31]. Equation (Equation 8) summarizes the operation of the SAT block.
(8)$$\begin{array}{c} y=x+SAT(Conv_{1\times 1}\left(GRN\left(Conv_{1\times 1}\left(DWConv_{7\times 7}\left(x\right)\right)\right)\right)), \end{array}$$
where $Conv_{k\times k}\left(\cdot \right)$ and $DWConv_{k\times k}\left(\cdot \right)$ denote convolution and depthwise convolution with a kernel size of k×k, respectively.

#### 2.3.3. Spatial Attention by Transformer (SAT) Module

Spatial attention is a mechanism in which the model emphasizes important features by multiplying feature elements by importance weights computed for each position. Lots of previous studies compute spatial attention weights by convolution [47,66]. However, such models have a limitation in that they estimate importance weights only from local contexts without considering the global context.

To overcome this limitation, we propose a novel SAT module that refers to the global context to compute spatial attention maps by applying a Transformer instead of convolution. In CSAT, such modification increases the number of parameters by only 1.8 K. However, the experimental results presented in Section 3.4 show that attention maps estimated from the global context can lead to higher performance than those estimated from local contexts in multiple pathological image analysis tasks.

One burden of applying a Transformer to a lightweight network is its computational and memory complexity, which scales as the square of the input resolution. To reduce the computational and memory overhead, we compute the attention map at a reduced resolution and then upsample the attention map to match the resolution of the feature maps.

Following Woo et al. [47], we first reduce the channel dimension by applying max-pooling and average-pooling in the channel direction as $x_{max}^{s}=P_{max}\left(\cdot \right)\in R^{1\times H\times W}$, $x_{avg}^{s}=P_{avg}\left(\cdot \right)\in R^{1\times H\times W}$ and concatenate the results as $\left[x_{max}^{s},x_{avg}^{s}\right]\in R^{2\times H\times W}$.

In general, each channel in the feature map represents a concept or object [42], and a pooling operation along the channel axis aggregates this information at each position. Then, we downsample the feature map using adaptive average pooling $P_{aap}\left(\cdot \right)$ with a fixed output resolution of $h^{\prime }\times w^{\prime }$, where $h^{\prime }<h$ and $w^{\prime }<w$. In this study, we set $h^{\prime }=w^{\prime }=7$. We apply an additional convolution layer to abstract the reduced feature map.
(9)$$\begin{array}{c} M_{S}=f_{spatial}\left(x\right)=Conv_{7x7}\left(P_{aap}\left(\left[P_{avg}\left(x\right);P_{max}\left(x\right)\right]\right)\right) \end{array}$$

The Transformer block takes as input the reduced feature map $M_{S}$ combined with positional encoding and produces an attention map at a resolution of $h^{\prime }\times w^{\prime }$. SAT applies relative positional encoding computed by the position encoding generator (PEG) [67]. PEG encodes positional information based on the local neighborhood of input tokens, which makes the model applicable to images of different resolutions.

The Transformer block computes an attention map from the global context at a reduced scale of $h^{\prime }\times w^{\prime }$ as Equation (Equation 10). Since we set $h^{\prime }$ and $w^{\prime }$ to small numbers, the additional overhead is minimal.
(10)$$\begin{array}{cc} Q,K,V & =Linear(M_{S}+PEG\left(M_{S}\right))\\ SA(Q,K,V) & =softmax(\frac{QK^{T}}{\sqrt{d}})V, \end{array}$$
where *Q*, *K*, and *V* are the query, key, and value vectors, *d* is feature dimension, and SA(·) denotes self-attention.

Finally, we upsample the attention map to the size of the input feature map through bilinear interpolation and multiply the upsampled attention map to the feature map as Equation (Equation 11).
(11)$$\begin{array}{cc} f_{att}\left(M_{S}\right) & =Upsample(SA(Q,K,V))\\ SAT(x) & =x\times f_{att}\left(f_{spatial}\left(x\right)\right). \end{array}$$

### 2.4. Evaluation Metrics and Experimental Environments

We evaluated the performance in classification and segmentation tasks using the metrics of accuracy and mean intersection over union (mIoU), respectively, which are computed as Equation (Equation 12).
(12)$$\begin{array}{cc} accuracy(\%) & =\frac{\#\;\mathrm{of correctly classified samples}}{\mathrm{total}\;\#\;\mathrm{of samples}}\times 100\%\\ mIoU & =\frac{1}{C}\sum_{i=1}^{C}\frac{TP_{i}}{TP_{i}+FP_{i}+FN_{i}} \end{array}$$
where *C*, TP, FP, and FN, respectively denote the number of classes, true positives, false positives, and false negatives. Accuracy indicates how accurately the model predicts the class of the input image across the entire dataset, while mIoU represents the average of the ratio of the intersection over the union between the predicted and ground truth regions across multiple classes, measuring the overlap between them.

We conducted experiments on a computer equipped with eight NVIDIA RTX A5000 GPUs, an Intel Xeon Gold 6226R CPU, and 540 GB of RAM. We built the software environment based on Ubuntu 20.04, PyTorch v2.0 [68], CUDA v11.1, and CuDNN v8.5. We counted the parameters and measured the amount of computation of the models in MACs using the THOP library [69]. For the downstream tasks, we only counted the parameters of the backbone because the parameters and MACs vary depending on the type of task. However, in Section 3.5, we compared the number of parameters of the entire model with those of the baseline models, including the task-specific head.

## 3. Results

In this section, we present the results of experiments to evaluate the performance and efficiency of the proposed RCKD and CSAT compared with the pretraining methods and model architectures proposed in previous studies.

### 3.1. Hyperparameter Search for Downstream Tasks

We first conducted preliminary experiments on the Lymph dataset to choose hyperparameters. We applied multiple candidate values for each hyperparameter and compared the performance. For the input resolution, we compared two candidates: 224×224 and 384×384. For the learning rate, we compared 0.001 and 0.0001. We also compared the performance of the models trained with and without freezing the positional encoding.

The results are presented in Figure 5. The best performance was achieved with an input resolution of 384×384, a learning rate of 0.0001, and a frozen positional encoder. We used these settings in all experiments. We fine-tuned the models with stochastic gradient descent (SGD) optimizer with a batch size of 64. We trained the models for a maximum of 200 epochs. However, we stopped training if the validation loss did not decrease for more than 20 epochs.

### 3.2. Performance in Downstream Tasks by Pretraining Method

We compared the performance of RCKD with three widely used pretraining methods: supervised pretraining (SPT) on the ImageNet dataset and two contrastive learning methods, Barlow Twins and MoCo. We pretrained four models for each of two distinct network architectures, ResNet18 and CSAT, employing four different pretraining methods. After transferring these pretrained models to the six downstream tasks listed in Table 1, we fine-tuned them on the target datasets. Then, we measured the performance of each model on the corresponding test sets. We also measured the performance of a model trained for the downstream tasks from random parameters without any pretraining. Figure 6 and Table 3 present the results.

RCKD significantly outperformed the three baseline pretraining methods. When applied to ResNet18, RCKD exhibited an average accuracy of 92.6% in the classification tasks, which is 7.6∼11% higher than the baseline pretraining methods. When applied to CSAT, the average accuracy of RCKD was 94.2%, which is 7.9∼25.5% higher than the baseline methods. In the segmentation tasks, RCKD showed average mIoUs of 0.665 and 0.673 when applied to ResNet18 and CSAT, respectively, which are 0.035∼0.046 and 0.002∼0.107 mIoUs higher than the baseline methods.

### 3.3. Performance in Downstream Tasks by Model Architecture

To evaluate the efficiency and effectiveness of the proposed CSAT, we conducted a comparative evaluation with two recently developed lightweight models, EfficientNet-B0 and ConvNextV2-Atto, which show outstanding performance with a small number of parameters. EfficientNet-B0 and ConvNextV2-Atto consist of 3.8 M and 3.2 M parameters, respectively, which are slightly larger than the size of CSAT with 2.8 M parameters.

Figure 7 and Table 4 present the evaluation results.

In the classification tasks, the CSAT pretrained by RCKD showed the best average accuracy of 94.2%. This model performed best for the BACH and BreakHis datasets. The supervised pretrained EfficientNet-B0 performed best on the CRC dataset, and the EfficientNet-B0 pretrained by RCKD performed best on the Lymph dataset. Despite having only 2.8 M parameters, which is only 73.6% of the parameters in EfficientNet-B0, CSAT showed slightly higher performance on average. Among the supervised pretrained baseline models, EfficientNet-B0 exhibited the highest average accuracy of 90.2%. CSAT pretrained by RCKD outperformed this model by 4%.

CSAT pretrained by RCKD also performed best in the segmentation tasks, showing an average mIoU of 0.673. However, on the BACH (WSI) dataset, the supervised pretrained CSAT exhibited the best performance. Both CSAT models outperformed EfficientNet-B0 and ConvNextV2-Atto on average.

### 3.4. Comparison with Previous Studies on Pathological Image Analysis

We compared the proposed methods with two recent studies on pathological image analysis, Riasatian et al. [70] and Ciga et al. [12]. Riasatian et al. [70] present KimiaNet pretrained by weakly supervised learning. Ciga et al. [12] apply SimCLR, a contrast learning method for pretraining. We also included the masked auto-encoder (MAE) [71] in the baseline models because, although it was not specialized for pathological image analysis, it has shown excellent performance in recent studies on computer vision. For Riasatian et al. [70] and Ciga et al. [12], we initialized the model by the pretrained parameters provided by the authors. However, because we were unable to find any pretrained MAE models specifically designed for pathological images, we only pretrained the MAE model using the TCGA dataset.

Figure 8 and Table 5 present the model architecture, pretraining method, the number of parameters, computational complexity in GMAC, and the performance of the models.

Despite the significantly smaller number of parameters and lower computational complexity compared to the baseline models, the proposed model performed best on five datasets among six downstream datasets and outperformed the baseline models on average in both classification and segmentation tasks. On the Lymph dataset, ResNet18 pretrained with SimCLR performed best. However, the proposed model showed a classification accuracy on the Lymph dataset only 0.8% lower than the best model [12] with only 26.4% of the parameters and 20.1% of the computation compared to the best model.

### 3.5. Evaluation of CSAT on ImageNet

Finally, we evaluated the performance of CSAT in general image classification using the ImageNet dataset. In this experiment, we compared CSAT with ResNet18 because ResNet18 is a lightweight model and widely used in computer vision. For an ablation study, we built two variants for both CSAT and ResNet18, one with the SAT module applied and one without. We trained the four models following USI [61], which is based on knowledge distillation and modern tricks. Figure 9 and Table 6 presents the results. With the SAT module, CSAT showed 2.4% higher classification accuracy than ResNet18 combined with the SAT module using 73.8% fewer parameters and 79.9% less computation. Without the SAT module, CSAT exhibited 2.7% higher accuracy than the vanilla ResNet18 model. The SAT module increased the accuracy of ResNet18 by 0.5% and that of CSAT by 0.2%.

### 3.6. Accuracy vs. Efficiency

In order to effectively analyze high-resolution pathological images in a general environment, not only performance but also computational and parameter efficiency are important. Therefore, we compared the accuracy, the amount of computation, and the number of parameters of models according to architecture and pretraining method. Figure 10 summarizes the results. The horizontal axis represents the amount of computation, while the vertical axis represents the average classification accuracy over the BACH (microscopy), CRC, BreakHis, and Lymph datasets. The color and size of each circle represent the pretraining methods and the number of parameters, respectively.

## 4. Discussion

As a transfer learning framework, RCKD has the following advantages. RCKD does not require labeled data for pretraining and thus enables the student model to learn a lot of knowledge from a large amount of unlabeled data. Furthermore, in RCKD, the student model learns not only from the data but also from the teacher model. This is an important advantage over conventional pretraining approaches where the model only learns from one knowledge source, the training data. Unlike conventional knowledge distillation, RCKD can transfer knowledge from a teacher model developed for a different task. Moreover, unlike conventional transfer learning methods, RCKD learns knowledge from the teacher model without transferring model parameters, so it is applicable even when the teacher and student models have different architectures.

On the other hand, RCKD shares the limitation of KD that the performance of the student model depends on that of the teacher model. The experimental results in Section 3 show that the nuclei segmentation model, StarDist, is strong enough to guide the training of the student model. However, to apply RCKD to other fields, it is necessary to search for teacher models that perform well in the areas where knowledge can be shared with the downstream task.

It is unclear why contrastive learning and MAE are less effective in pathological image analysis than in other computer vision fields. One possible reason is the unique characteristics of pathological images that differ from general images. While general images mainly comprise objects with consistent large-scale shapes, pathological images are composed of tissues with irregular sizes and shapes. We suspect that conventional pretraining techniques prioritize global patterns over local details, despite the latter’s significance in the analysis of pathological images.

## 5. Conclusions

Major challenges in pathological image analysis include the scarcity of labeled data and the characteristics of pathological images significantly different from ordinary images, which limits the effect of conventional transfer learning techniques. To overcome these limitations, we proposed a novel Response-based Cross-task Knowledge Distillation (RCKD) framework that learns knowledge from unlabeled pathological images guided by a teacher model developed for a different task. In experiments, RCKD outperformed supervised pretraining and contrastive learning by large margins. RCKD has additional advantages in that it does not require manual labeling and can learn knowledge from a teacher model with different architecture or target tasks. We also proposed the Convolutional neural network with Spatial Attention by Transformers (CSAT), a lightweight architecture for the processing of high-resolution pathological images, such as pathological images. CSAT outperformed ResNet18 on ImageNet by a large margin. The CSAT pretrained by RCKD exhibited average performances of 94.2% in classification tasks and 0.673 mIoU in segmentation tasks, which are 3.9∼14% and 0.047∼0.103 mIoU higher than recent pathological image analysis models, respectively. We expect that the results of this study will improve the performance and efficiency of deep learning-based pathological image analysis models, thereby accelerating the development of key techniques for AI-assisted, or fully automated diagnosis.

## Figures and Tables

**Figure 1 bioengineering-10-01279-f001:**
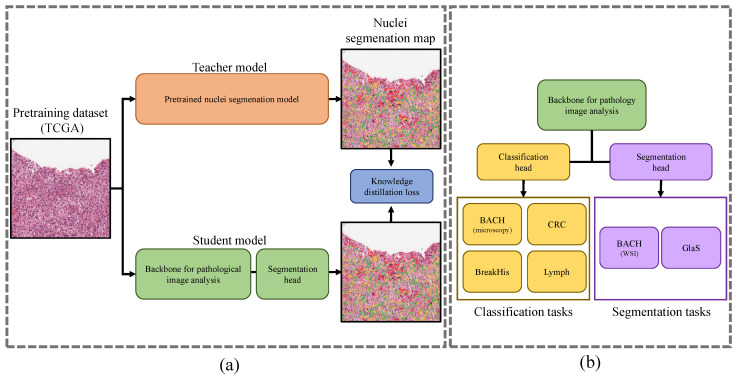
The overall procedure of RCKD. (**a**) Pretraining from unlabeled pathological images. (**b**) Fine-tuning for downstream tasks.

**Figure 2 bioengineering-10-01279-f002:**
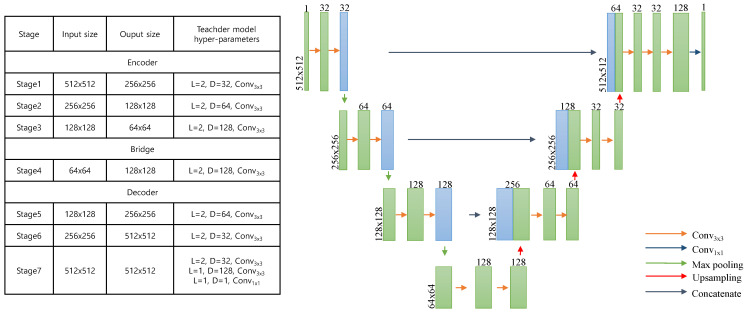
The structure and hyperparameters of the teacher model, StarDist. The kernel size and stride of the Max pooling are two in all layers. For upsampling, StarDist utilizes bilinear interpolation with an upsampling ratio of two.

**Figure 3 bioengineering-10-01279-f003:**
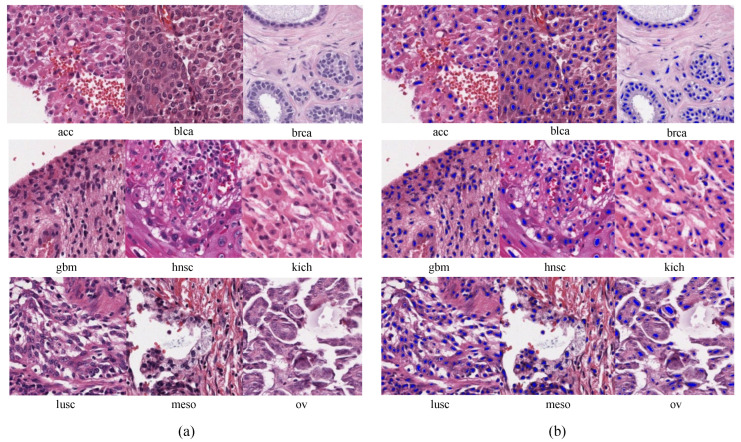
Pathological images in the TCGA dataset by organ (**a**) and the corresponding binary nuclei segmentation maps estimated by the teacher model, StarDist (**b**). The blue color indicates the estimated nuclei regions. (acc: adrenocortical carcinoma, blca: bladder urothelial carcinoma, brca: breast invasive carcinoma, gbm: glioblastoma multiforme, hnsc: head and neck squamous cell carcinoma, kich: kidney chromophobe, lusc: lung squamous cell carcinoma, meso: mesothelioma, and ov: ovarian serous cystadenocarcinoma).

**Figure 4 bioengineering-10-01279-f004:**
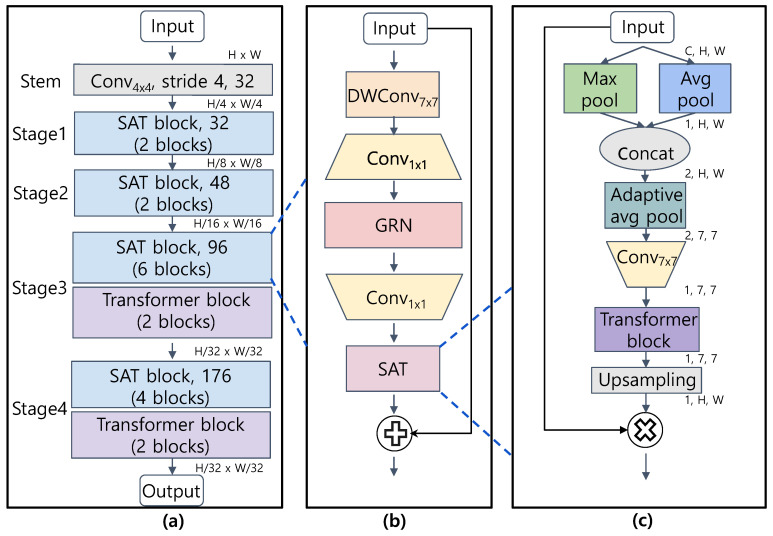
The architecture of CSAT. (**a**) overall structure, (**b**) SAT block, and (**c**) SAT module. ${\mathrm{Conv}}_{k\times k}$, ${\mathrm{DWConv}}_{k\times k}$ represent the convolution and depthwise convolution with a kernel size of k×k, respectively. *C*, *H*, and *W* denote the channel, height, and width of the input image, respectively.

**Figure 5 bioengineering-10-01279-f005:**
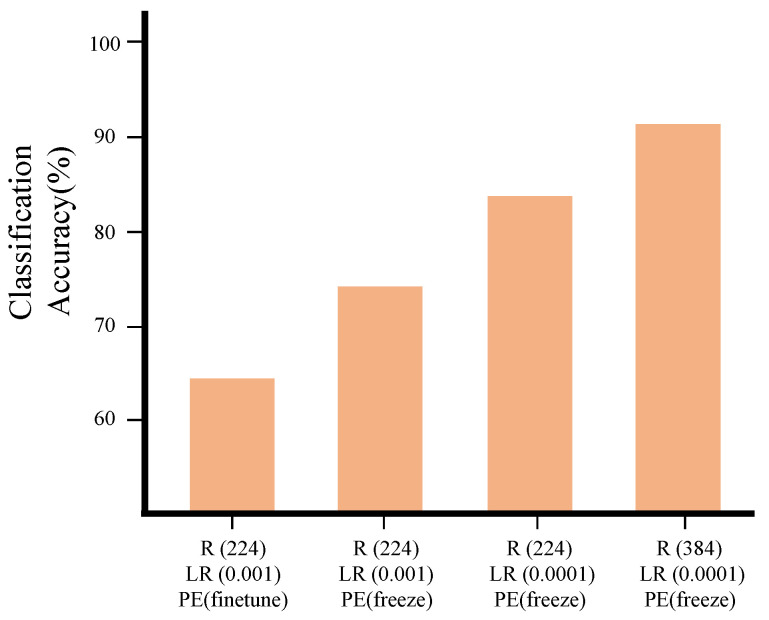
Classification accuracy of CSAT models on the Lymph dataset according to hyperparameters. The graph displays the average result of five experiments conducted with different data splits. R, LR, and PE denote the resolution of the input image, the learning rate, and the weight used in the positional encoding, respectively.

**Figure 6 bioengineering-10-01279-f006:**
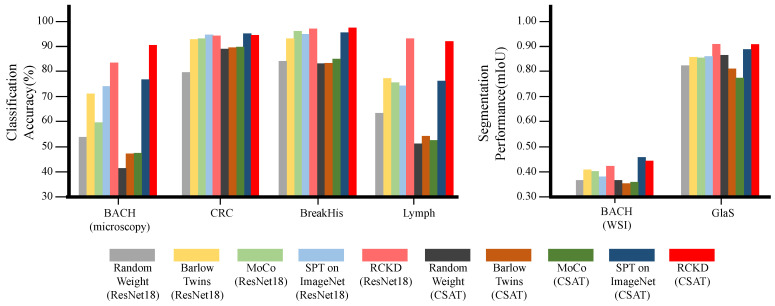
The performance of CSAT and ResNet18 by pretraining methods for four classification tasks (BACH (microscopy), CRC, BreakHis, and Lymph) and two segmentation tasks (BACH (WSI) and GlaS).

**Figure 7 bioengineering-10-01279-f007:**
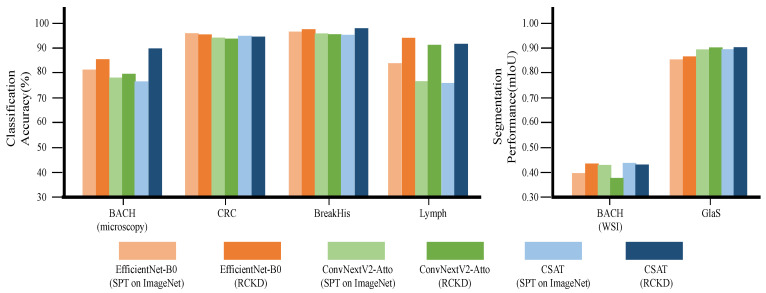
The performance of CSAT and recent lightweight models on six pathological image analysis tasks. (SPT denotes supervised pretraining).

**Figure 8 bioengineering-10-01279-f008:**
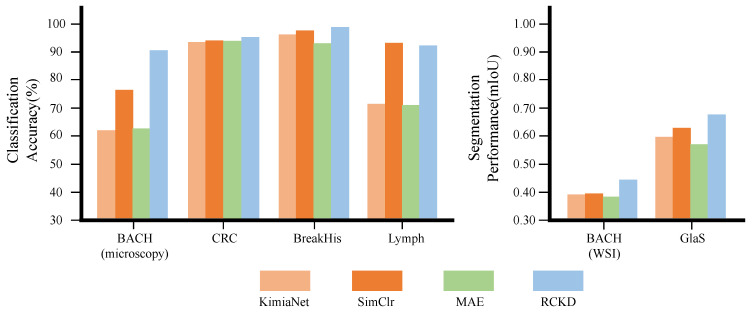
The performance of CSAT pretrained by RCKD compared with three previous studies on six pathological image analysis tasks.

**Figure 9 bioengineering-10-01279-f009:**
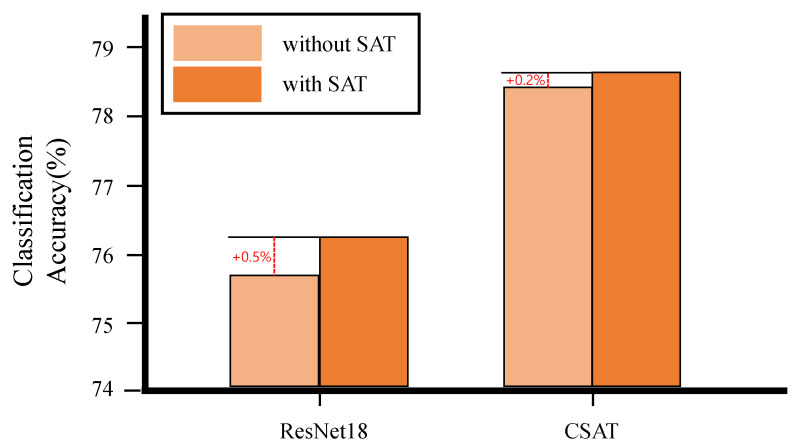
The performance of CSAT and ResNet18 on the ImageNet dataset.

**Figure 10 bioengineering-10-01279-f010:**
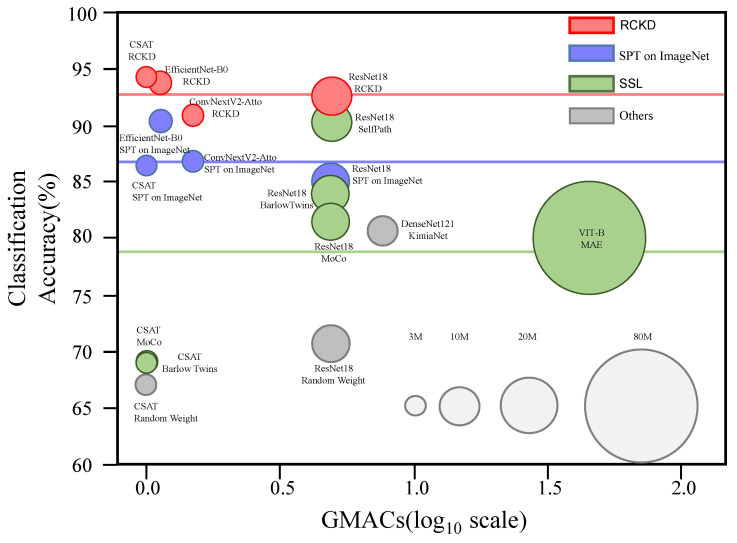
A ball chart displaying the average classification accuracy according to model architecture and pretraining method. The vertical axis represents the accuracy averaged for four classification tasks listed in Table 1. The horizontal axis represents the computational complexity in GMAC. The color and size of each circle represent the pretraining methods and the number of parameters, respectively. The average classification accuracy of the models pretrained by RCKD, supervised pretraining (SPT) on ImageNet, and self-supervised learning (SSL) on pathological images are 92.8% (red line), 87% (blue line), and 78.9% (green line), respectively.

**Table 1 bioengineering-10-01279-t001:** Pathological image datasets for target tasks. (BACH: the BreAst Cancer Histology dataset, CRC: ColoRectal Cancer dataset, BreakHis: the Breast cancer Histopathological dataset, GlaS: the Gland Segmentation in colon histology images dataset).

Datasets	Tasks	Classes	Data Size	Number of Patches [Train, Validation, Test]	Magnification/ Patch Size
BACH [50] (microscopy)	Breast cancer subtype classification	Normal, Benign, In situ carcinoma, Invasive carcinoma	400 patches	400 [240, 80, 80]	20×/ 2048 × 1536
CRC [51]	Colorectal cancer and normal tissue classification	Adipose, Background, Debris, Lymphocytes, Mucus, Smooth muscle, Normal colon mucosa, Cancer-associated stroma, Colorectal adenocarcinoma epithelium	107,180 patches	107,180 [80,000, 20,000, 7180]	20×/ 224 × 224
BreakHis [52]	Malignant and benign tissue classification in breast cancer	Benign tumors, Malignant tumors	7909 patches	7909 [4745, 1582, 1582]	4×, 10×, 20×, 40×/ 700 × 460
Lymph [53]	Malignant lymph node cancer classification	Chronic lymphocytic leukemia, Follicular lymphoma, Mantle cell lymphoma	374 patches	374 [224, 75, 75]	40×/ 1388 × 1040
BACH [50] (WSI)	Breast cancer subtype segmentation	Normal, Benign, In situ carcinoma, Invasive carcinoma	10 WSIs	4483 [2388, 1214, 881]	10×/ 1024 × 1024
GlaS [54]	Gland segmentation	Benign gland, Malignant gland	165 patches	165 [68, 17, 80]	20×/ 775 × 522

**Table 2 bioengineering-10-01279-t002:** The structure and hyper-parameters of CSAT. *L* denotes the number of blocks and *D* denotes the number of channels.

Stages	Input Size	Output Size	CSAT Hyper-Parameters
Stem	H, W	H/4, W/4	L = 1, D = 32, Convolution block
Stage1	H/4, W/4	H/8, W/8	L = 2, D = 32, SAT block
Stage2	H/8, W/8	H/16, W/16	L = 2, D = 48, SAT block
Stage3	H/16, W/16	H/32, W/32	L = 6, D = 96, SAT block L = 2, D = 96, Transformer block
Stage4	H/32, W/32	H/32, W/32	L = 4, D = 176, SAT block L = 2, D = 176, Transformer block

**Table 3 bioengineering-10-01279-t003:** The performance of CSAT and ResNet18 by pretraining methods on six pathological image analysis tasks. The boldface indicates the best performance. (Params, mIoU, and GMAC denote the number of parameters, mean intersection over union, and giga multiply-accumulate operations, respectively).

Model	Params	GMAC	Pretraining Methods	Classification Accuracy (%)					Segmentation Performance (mIoU)		
				BACH (Microscopy)	CRC	BreakHis	Lymph	Average Accuracy	BACH (WSI)	GlaS	Average mIoU
ResNet18	10.6 M	5.35	Random Weight	53.7	80.2	85.0	63.7	70.6	0.355	0.83	0.592
	10.6 M	5.35	Barlow Twins	71.4	93.7	93.8	77.8	84.1	0.40	0.861	0.630
	10.6 M	5.35	MoCo	59.7	93.8	96.9	76.0	81.6	0.391	0.864	0.627
	10.6 M	5.35	SPT on ImageNet	74.4	**95.5**	95.5	74.6	85	0.373	0.866	0.619
	10.6 M	5.35	RCKD	**83.9**	95.0	**98.0**	**93.8**	**92.6**	**0.415**	**0.915**	**0.665**
CSAT	2.8 M	1.08	Random Weight	41.3	89.6	83.7	51.4	66.5	0.355	0.872	0.613
	2.8 M	1.08	Barlow Twins	47.0	90.2	83.7	53.9	68.7	0.342	0.81	0.576
	2.8 M	1.08	MoCo	47.1	90.6	85.5	52.6	68.9	0.35	0.783	0.566
	2.8 M	1.08	SPT on ImageNet	77.0	**95.8**	96.1	76.5	86.3	**0.441**	0.902	0.671
	2.8 M	1.08	RCKD	**90.6**	95.3	**98.6**	**92.5**	**94.2**	0.435	**0.912**	**0.673**

**Table 4 bioengineering-10-01279-t004:** The performance of CSAT and recent lightweight models on six pathological image analysis tasks. The boldface indicates the best performance. (Params, mIoU, and GMAC denote the number of parameters, mean intersection over union, giga multiply-accumulate operations, respectively).

Model	Params	GMAC	Pretraining Methods	Classification Accuracy (%)					Segmentation Performance (mIoU)		
				BACH (Microscopy)	CRC	BreakHis	Lymph	Average Accuracy	BACH (WSI)	GlaS	Average mIoU
EfficientNet-B0	3.8 M	1.21	SPT on ImageNet	82.2	**96.8**	97.4	84.7	90.2	0.399	0.861	0.630
EfficientNet-B0	3.8 M	1.21	RCKD	85.9	96.2	98.3	**94.9**	93.8	0.438	0.873	0.655
ConvNextV2-Atto	3.2 M	1.60	SPT on ImageNet	78.6	95.1	96.6	77.2	86.8	0.434	0.901	0.667
ConvNextV2-Atto	3.2 M	1.60	RCKD	80.4	94.9	96.4	91.9	90.9	0.38	0.91	0.645
CSAT	2.8 M	1.08	SPT on ImageNet	77.0	95.8	96.1	76.5	86.3	**0.441**	0.902	0.671
CSAT	2.8 M	1.08	RCKD	**90.6**	95.3	**98.6**	92.5	**94.2**	0.435	**0.912**	**0.673**

**Table 5 bioengineering-10-01279-t005:** The performance of CSAT pretrained by RCKD compared with three previous studies on six pathological image analysis tasks. The boldface indicates the best performance. (Params, mIoU, and GMAC denote the number of parameters, mean intersection over union, and giga multiply-accumulate operations, respectively).

Pretraining Methods	Model	Params	GMAC	Classification Accuracy (%)					Segmentation Performance (mIoU)		
				BACH (Microscopy)	CRC	BreakHis	Lymph	Average Accuracy	BACH (WSI)	GlaS	Average mIoU
KimiaNet [70]	DensNet121	6.6M	8.51	61.3	93.8	96.3	71.5	80.7	0.385	0.804	0.594
SimCLR [12]	ResNet18	10.6M	5.35	76.2	94.2	97.7	**93.3**	90.3	0.386	0.866	0.626
MAE [71]	ViT-B	81.6M	49.3	62.3	94.1	93.1	71.3	80.2	0.378	0.763	0.570
RCKD	CSAT	2.8M	1.08	**90.6**	**95.3**	**98.6**	92.5	**94.2**	**0.435**	**0.912**	**0.673**

**Table 6 bioengineering-10-01279-t006:** The performance of CSAT and ResNet18 on the ImageNet dataset. (Params and GMAC represent the number of parameters and giga multiply-accumulate operations, respectively).

Model	SAT Module	Params	GMAC	Classification Accuracy (%)
ResNet18	X	11,689,512	5.35	75.7
ResNet18	O	11,690,544	5.35	76.2
CSAT	X	3,063,272	1.08	78.4
CSAT	O	3,065,078	1.08	78.6

## Data Availability

The code in this article cannot be published due to privacy and can be obtained from the corresponding author upon reasonable request.

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
