# Peer review of "RCKD: Response-Based Cross-Task Knowledge Distillation for Pathological Image Analysis"

_bioengineering, 2023, doi:10.3390/bioengineering10111279_

Round 1

Reviewer 1 Report

Comments and Suggestions for Authors

This study presents a deep learning technology to improve pathological image analysis through a novel <teacher-student> learning scheme, which overall has great potential due to its ability to be applied to unlabelled datasets. Examples of large pathological image databases demonstrate the potential and real-life application of the technique.

Although the method has the potential to improve the traditional analysis of pathological images, this article needs careful revision according to the revision comments. They mainly reveal problems with the correct presentation of scientific information according to guidelines.

Detailed list of revision remarks follow:

1.       The term "pathology image" is noticed several times in a single sentence, as well as in consecutive sentences. Obviously, such duplication implies repetition of ideas and leads to misunderstanding. Authors should reconfigure their statements into complete sentences without duplication. Furthermore, note that the correct English term is “pathological image”.

2.        A problem with figures numbering and placement is noticed! As written in the journal’s guidelines: “All Figures, Schemes and Tables should be inserted into the main text close to their first citation and must be numbered following their number of appearance (Figure 1, Scheme I, Figure 2, Scheme II, Table 1, etc.).”. For example, Figure 1 is placed in Introduction but referenced in the text in section 5. Experiments. Instead, Fig.3 is the first one referenced in the text (page 3) but appear in page 7. The figure (named algorithm) in page 8 is NOT numbered! Correct all these and other such inconsistencies.

3.       If any figure is replicated from a third party source, then permission should be asked and disclosed.

4.       Ln 70-105: INCORRECT disclosure of methodological details in Introduction! Instead, the AIMS of the study are missing They should be present after “Related works” and should should be a brief definition of the still unsolved problems, the presented novel ideas and expected clinical benefits. Do not go into methodological details, which are incomprehensive at this early stage of the article. Furthermore, “the major contributions of this study” (Ln 97-105) are relevant for CONCLUSIONS but not for aims of the study, which should be at the 'idea level', but not to reporting methods and performance values.

5.       Ln 106-108: This paragraph contains irrelevant information that conflicts with standard article formatting as defined in the journal's guidelines (https://www.mdpi.com/journal/bioengineering/instructions): Research manuscripts must include sections: 1. Introduction, 2. Materials and Methods, 3. Results, 4. Discussion, 5. Conclusions. Therefore, the manuscript should be formatted according to the guidelines and the reader does NOT need a guide "How to Read the Paper". In this way, section “2. RelatedWorks” must be embedded in section Introduction; Sections “3. Response-based cross-task knowledge distillation must” and “4. Convolutional neural network with spatial attention by transformer” must be embedded in “2. Materials and Methods”; section “5. Experiments” must be renamed ‘Results”; Section “Conclusions and Limitations” must be split in different sections as their content is completely different.

6.       Missing section Discussion!!! Authors should provide a clear analysis of the pros and cons of this study, as well as a comparison with the results of other similar studies in the field.

7.       Ln 200-211: “RCKD is a pretraining framework…” -> Missing information about the network source The authors go into general information, but the Methods section needs an introduction that references all model sources and input data sources. The sources and connection scheme of “nuclei segmentation maps”, “student model”, “segmentation head”, “pretraining network” should be presented in sufficient detail.

8.       Methods: Missing architecture of the teacher and student models.

9.       Figure 2 (and all others with this problem): All abbreviations must be explained in the caption so that this figure can be read outside the main text.

10.    Ln 264: “a base learning rate of 0” -> Check the learning rate as it should be >0

11.    Table 1: Provide references for the Datasets (1st column). Explain all heading row abbreviations in the caption. Make sure they directly correspond to the terms defined in the main text. The sizes of the databases are missing.

12.    Use the proper format for third-level headings as defined in the journal template. The use of bolded text is not appropriate.

13.    Figure 4 contains numerical results, therefore, it is not appropriate in section “2. Materials in Methods”. The figure must be placed and explained as appropriate in section “3. Results”.

14.    Ln 396-405 must be fully rewritten according to the new structure of the article.

15.    Page 13: Footer notes are not permitted in the journal. All sources (including URLs) must be properly cited in section References.

16.    Figures 6,7,8,9: The accuracy metrics must be labeled. Make sure that the computation of all accuracy metrics is defined in section “2. Materials and Methods”, otherwise the graphs are not interpretable.

17.    Table 6 reports “Accuracy” but other Tables 3,4,5 (and figures) report “average” performance. Make sure that all performance metrics are homogeneous and well explained so that there is a clear transfer of information between the figures and tables.

Comments on the Quality of English Language

See revision comment 1.

Author Response

We are grateful to the reviewers for their time and effort spent reviewing our manuscript. We have carefully considered all of the reviewer's comments and suggestions and have made appropriate revisions to the manuscript. For the convenience of the reviewers, all the revised portions in the manuscript have been highlighted in blue. Our responses to each comment are outlined below. We believe the revised manuscript and our response will adequately address any concerns raised by the reviewers.

1) The term "pathology image" is noticed several times in a single sentence, as well as in consecutive sentences. Obviously, such duplication implies repetition of ideas and leads to misunderstanding. Authors should reconfigure their statements into complete sentences without duplication. Furthermore, note that the correct English term is “pathological image”.

=> We thoroughly checked and revised the sentences that include ‘pathology image’ multiple times (Line 21-66). We also replaced ‘pathology image’ with ‘pathological image’. (We didn’t mark them blue because this term was used in too many places.)

2)  A problem with figures numbering and placement is noticed! As written in the journal’s guidelines: “All Figures, Schemes and Tables should be inserted into the main text close to their first citation and must be numbered following their number of appearance (Figure 1, Scheme I, Figure 2, Scheme II, Table 1, etc.).”. For example, Figure 1 is placed in Introduction but referenced in the text in section 5. Experiments. Instead, Fig.3 is the first one referenced in the text (page 3) but appear in page 7. The figure (named algorithm) in page 8 is NOT numbered! Correct all these and other such inconsistencies.

=> We rearranged the numbering, order, and location of figures and tables according to the comment. We also re-labeled Algorithms 1 and 2 as Figure 4 and 5 respectively. (Line 314-315, 339-340)

3) If any figure is replicated from a third party source, then permission should be asked and disclosed.

=> We drew all figures and the manuscript does not contain any figure from a third party source.

4) Ln 70-105: INCORRECT disclosure of methodological details in Introduction! Instead, the AIMS of the study are missing They should be present after “Related works” and should be a brief definition of the still unsolved problems, the presented novel ideas and expected clinical benefits. Do not go into methodological details, which are incomprehensive at this early stage of the article. Furthermore, “the major contributions of this study” (Ln 97-105) are relevant for CONCLUSIONS but not for aims of the study, which should be at the 'idea level', but not to reporting methods and performance values.

=> We have reorganized the structure according to the comment. Specifically, we described the aim of our study in the Introduction under Section 1.3 (Lines 159-172) and moved the methodological details to Section 3.  We deleted 'contributes' from Introduction and modified the content of conclusion.

5) Ln 106-108: This paragraph contains irrelevant information that conflicts with standard article formatting as defined in the journal's guidelines (https://www.mdpi.com/journal/bioengineering/instructions): Research manuscripts must include sections: 1. Introduction, 2. Materials and Methods, 3. Results, 4. Discussion, 5. Conclusions. Therefore, the manuscript should be formatted according to the guidelines and the reader does NOT need a guide "How to Read the Paper". In this way, section “2. RelatedWorks” must be embedded in section Introduction; Sections “3. Response-based cross-task knowledge distillation must” and “4. Convolutional neural network with spatial attention by transformer” must be embedded in “2. Materials and Methods”; section “5. Experiments” must be renamed ‘Results”; Section “Conclusions and Limitations” must be split in different sections as their content is completely different.

=> We modified the entire structure of the manuscript. The structure of the revised manuscript follows the journal’s guidelines.

6) Missing section Discussion!!! Authors should provide a clear analysis of the pros and cons of this study, as well as a comparison with the results of other similar studies in the field.

=> We added Discussion (Line 522-545) section in the revised manuscript.

7) Ln 200-211: “RCKD is a pretraining framework…” Missing information about the network source. The authors go into general information, but the Methods section needs an introduction that references all model sources and input data sources. The sources and connection scheme of “nuclei segmentation maps”, “student model”, “segmentation head”, “pretraining network” should be presented in sufficient detail.

⇒ We added the detailed description of RCKD in Section 2.2 and source of the networks, as follows.

  • Nuclei segmentation model (the teacher model): (Line 292-297)

“For the teacher model f_teacher(·), we used StarDist [56] pretrained on the MoNuSeg2018 [57] and TNBC [58] datasets. StarDist won the CoNIC challenge in 2022 [59]. StarDist segments nuclei regions using a U-Net [60] based model and represents them as star convex polygons. The structure of StarDist are presented in Fig. 2. Fig. 3 displays the pathological image samples used for pretraining and the pseudo label N(x) estimated by StarDist.”

  • The student model and segmentation head: We used CSAT as the student model, which is described in Subsection 2.3. We implemented the segmentation head using the decoder of U-Net. We specified this in Line 300-302 as follows:

“In this study, we implemented the student model using a novel lightweight network, CSAT, described in Section 2.3, and the segmentation head using the U-Net decoder.”

  • We understood ‘pretraining network’ to be ‘pretrained network’, which is the student model described above.
  • ‘Nuclei segmentation maps’: A nuclei segmentation map is a 2D map where each element is the probability that a pixel belongs to a nuclei region. We changed the term 'Nuclei Segmentation Map' to 'Binary Nuclei Segmentation Map' to clarify its meaning. We explained it in Line 287-290 and Eq. 1-2 in detail.

“y∈R^{HxW} is a probability map predicted by f_teacher(·), where y_ij is the estimated probability that a pixel x_ij belongs to a nuclei region. Then, we convert y into a binary segmentation map N(x) ∈ {0, 1}^{HxW} with a threshold value α, as Eq. 2.”

8) Methods: Missing architecture of the teacher and student models.

=> We added an overview of teacher model in Figure 2 (Line 295-296). The architecture details of the student model, CSAT, are presented in Subsection 2.3, including Fig. 6 and Table 2 (Line 362-363).

9) Figure 2 (and all others with this problem): All abbreviations must be explained in the caption so that this figure can be read outside the main text.

=> We explained all abbreviations in the captions of all figures and tables.

10) Ln 264: “a base learning rate of 0” -> Check the learning rate as it should be >0.

=> The LARS optimizer automatically adjusts the learning rate for each layer based on the ratio of the norm of the gradient to the norm of the layer weights. Even if the initial learning rate is set to 0, training will proceed. We have added a detailed explanation about this and references to the code we referred to in the revised manuscript. We explained this in Line 308-312 as follows:

“LARS automatically adjusts the learning rate for each layer, allowing training to proceed even when the initial learning rate is set to zero.”

11) Table 1: Provide references for the Datasets (1st column). Explain all heading row abbreviations in the caption. Make sure they directly correspond to the terms defined in the main text. The sizes of the databases are missing.

=> We added the size (the number of WSIs or patches depending on the dataset) of the datasets and cited the source in Table 1 (Line 240-241). We also explained the full names of the abbreviations in the caption. We described the downstream datasets in detail in Subsection 2.1.2 (Line 195-244).

12) Use the proper format for third-level headings as defined in the journal template. The use of bolded text is not appropriate.

=> We revised the format of headings to follow the journal format.

13) Figure 4 contains numerical results, therefore, it is not appropriate in section “2. Materials in Methods”. The figure must be placed and explained as appropriate in section “3. Results”.

=> We moved Figure 4 to the Result section (Section 3.1)  and renumbered it to Figure 7 (Line 443-444).

14) Ln 396-405 must be fully rewritten according to the new structure of the article.

=> We rewrote Lines 396-405 of the previous manuscript in Subsection 2.4 (Line 422-429) of the revised manuscript.

15) Page 13: Footer notes are not permitted in the journal. All sources (including URLs) must be properly cited in section References.

=> We moved the source of the THOP library from footnote to Reference [69].

16) Figures 6,7,8,9: The accuracy metrics must be labeled. Make sure that the computation of all accuracy metrics is defined in section “2. Materials and Methods”, otherwise the graphs are not interpretable.

=> We specified accuracy metrics in all figures presenting results and presented the formula of the metrics as Eq. 12 in Subsection 2.4 (Line 414-421).

17) Table 6 reports “Accuracy” but other Tables 3,4,5 (and figures) report “average” performance. Make sure that all performance metrics are homogeneous and well explained so that there is a clear transfer of information between the figures and tables.

=> We kept the metrics consistent: we used accuracy for all classification datasets and mIoU for all segmentation datasets. In Table 6 (Line : 511-512), we presented only classification accuracies because it reports performance on a single classification dataset, ImageNet. However, Tables 3-5 (Line 455-456, 474-475, 497-498) present the results on multiple downstream datasets and therefore present the accuracy on each classification datasets and mIoU for each segmentation tasks. In addition, we have added the average performance for each task type (classification or segmentation) to make it easier to compare the overall performance of the models and pretraining algorithms.

Reviewer 2 Report

Comments and Suggestions for Authors

The manuscript presents a novel approach for improved pathology image analysis that involves a Response-based Cross-task Knowledge Distillation (RCKD) framework for learning knowledge from unlabeled pathology image data and a convolutional neural network with spatial attention by transformer (CSAT) that is trained for classification of pathology images via RCKD. The presented methodology is well described and achieves promising results. The language is fine, which makes the manuscript easy for reading and understanding.

I have the following remarks that should be addressed before publication:

1)      What are the defined classes in the classification task? I recommend the authors to extend the description of the used datasets in Table 1, so that the reader  becomes familiar with the classification task. The authors have written: “In this study, we fine-tuned the pretrained model for four classification tasks and two segmentation tasks”. The authors should provide additional information about the classification and segmentation tasks, since they are not evident from Table 1.

In section ‘Experiments’, the authors should present equations for the calculation of “Classification accuracy” and “Segmentation performance”. What are the target classes and segments?

Author Response

We are grateful to the reviewers for their time and effort spent reviewing our manuscript. We have carefully considered all of the reviewer's comments and suggestions and have made appropriate revisions to the manuscript. For the convenience of the reviewers, all the revised portions in the manuscript have been highlighted in blue. Our responses to each comment are outlined below. We believe the revised manuscript and our response will adequately address any concerns raised by the reviewers.

1)  What are the defined classes in the classification task? I recommend the authors to extend the description of the used datasets in Table 1, so that the reader  becomes familiar with the classification task. The authors have written: “In this study, we fine-tuned the pretrained model for four classification tasks and two segmentation tasks”. The authors should provide additional information about the classification and segmentation tasks, since they are not evident from Table 1.

=> We added the target classes of each classification tasks in Table 1 (Line 240-241) and explained the downstream datasets in Subsection 2.1.2 (Line : 195-244).

2) In section ‘Experiments’, the authors should present equations for the calculation of “Classification accuracy” and “Segmentation performance”. What are the target classes and segments?

=> We presented the evaluation metrics for classification and segmentation tasks in Eq. (12) and explained them in Subsection 2.4 (Line 414-421).

“We evaluated the performance in classification and segmentation tasks using the metrics of accuracy and mean intersection over union (mIoU), respectively, which are computed as Eq. 12.

where C, TP, FP, and FN respectively denote the number of classes, true positives,  false positives, and false negatives. Accuracy indicates how accurately the model predicts the class of the input image across the entire dataset, while mIoU represents the average of the ratio of the intersection over the union between the predicted and ground truth regions across multiple classes, measuring the overlap between them.”

Round 2

Reviewer 1 Report

Comments and Suggestions for Authors

The authors have greatly improved their manuscript and satisfactorily responded to all revision comments. In my opinion, the current version of the manuscript conforms to the standard rules for the formatting and content of scientific articles. May be published as is.

A note to the authors: Ln 21-22: "Pathological image analysis aims to extract useful information from pathological images commonly acquired through a whole slide scanner or camera." -> This is an ambiguity. Pathological image analysis is applied to diagnostic images obtained from the general population, which includes healthy subjects from whom non-pathological images were obtained. I would say that pathological image analysis looks for the presence and type of pathology, but the image itself is not necessarily pathological.